# Assessment of *Ocimum basilicum* Essential Oil Anti-Insect Activity and Antimicrobial Protection in Fruit and Vegetable Quality

**DOI:** 10.3390/plants11081030

**Published:** 2022-04-10

**Authors:** Miroslava Kačániová, Lucia Galovičová, Petra Borotová, Nenad L. Vukovic, Milena Vukic, Simona Kunová, Pavel Hanus, Ladislav Bakay, Edyta Zagrobelna, Maciej Kluz, Przemysław Łukasz Kowalczewski

**Affiliations:** 1Institute of Horticulture, Faculty of Horticulture and Landscape Engineering, Slovak University of Agriculture, Tr. A. Hlinku 2, 94976 Nitra, Slovakia; l.galovicova95@gmail.com; 2Department of Bioenergy, Food Technology and Microbiology, Institute of Food Technology and Nutrition, University of Rzeszow, 4 Zelwerowicza St., 35601 Rzeszow, Poland; ezagrobelna@ur.edu.pl (E.Z.); kluczyk82@op.pl (M.K.); 3AgroBioTech Research Centre, Slovak University of Agriculture, Tr. A. Hlinku 2, 94976 Nitra, Slovakia; petra.borotova@uniag.sk; 4Institute of Applied Biology, Faculty of Biotechnology and Food Sciences, Slovak University of Agriculture in Nitra, Tr. A. Hlinku 2, 94976 Nitra, Slovakia; 5Department of Chemistry, Faculty of Science, University of Kragujevac, 34000 Kragujevac, Serbia; nvchem@yahoo.com (N.L.V.); milena.vukic@pmf.kg.ac.rs (M.V.); 6Institute of Food Sciences, Faculty of Biotechnology and Food Sciences, Slovak University of Agriculture, Tr. A. Hlinku 2, 94976 Nitra, Slovakia; simona.kunova@uniag.sk; 7Department of Food Technology and Human Nutrition, Institute of Food and Nutrition Technology, University of Rzeszow, 35959 Rzeszow, Poland; hanuspawel@gmail.com; 8Institute of Landscape Architecture, Faculty of Horticulture and Landscape Engineering, Slovak University of Agriculture, Tr. A. Hlinku 2, 94976 Nitra, Slovakia; ladislav.bakay@uniag.sk; 9Department of Food Technology of Plant Origin, Poznań University of Life Sciences, 31 Wojska Polskiego St., 60624 Poznań, Poland

**Keywords:** basil essential oil, antimicrobial activity, in vitro, in situ, bacteria, yeast, fruits, vegetables

## Abstract

Basil (*Ocimum basilicum*) is a commonly used herb; it also contains essential oils and other valuable compounds. The basil oil obtained has a pleasant aroma, but also a broad spectrum of biological activity. This work reports on the chemical composition, antioxidant, antimicrobial and anti-insect activity in vitro and in situ of *Ocimum basilicum* essential oil (OBEO) obtained by steam distillation of fresh flowering plants. Gas chromatography–mass spectrometry, DPPH, agar and disc diffusion and vapor phase methods were used to analyze the OBEO properties. The analysis of the chemical composition of OBEO showed that its main components were methyl chavicol (88.6%), 1,8-cineole (4.2%) and α-trans-bergamotene (1.7%). A strong antioxidant effect was demonstrated at the level of 77.3%. The analysis of antimicrobial properties showed that OBEO exerts variable strength of inhibiting activity against various groups of microorganisms. The growth inhibition zones ranged from 9.67 to 15.33 mm in Gram-positive (G^+^) and Gram-negative (G^−^) bacteria and from 5.33 to 7.33 mm in yeast. The lowest measured minimal inhibition concentration (MIC) was 3.21 µL/mL against Gram-negative *Azotobacter chrococcum* and Gram-positive *Micrococcus luteus*. The antimicrobial activity of in situ vapor phase of OBEO was also confirmed on apples, pears, potatoes and kohlrabi. The highest insecticidal activity against *Pyrrhocoris*
*apterus*, observed at the concentration of 100%, caused the death of 80% of individuals. Due to its broad spectrum of activity, OBEO seems an ideal candidate for preserving fruit and vegetables.

## 1. Introduction

Conventional antimicrobials can cause potential ecotoxicological risks and be hazardous to a wide range of non-target organisms in systems because they impact basic biological processes that are not unique to microorganisms [1]. The growing number of organic farms producing both fruit and vegetables creates a demand for organic compounds to control plant pathogens. Adapted as a worldwide agricultural strategy, integrated pest management (IPM) relies on the reduced use of pesticides [2,3]. Among the new sources of compounds used in organic agriculture, plant-derived compounds deserve special attention. The widely reported biological activity of essential oils indicates that they can play a role of compounds that control pests and plant pathogens [4,5]. The insecticidal, bactericidal, fungicidal and viricidal activity of the essential oils is due to the presence of various functional groups, such as alcohols, aldehydes, phenols, terpenes, ketones and other antimicrobial compounds [6,7]. However, the use of excessively high doses of essential oils (EOs) may be toxic for plants [8,9].

One of the most frequently mentioned factors causing severe losses of fruit and vegetables after their harvest is contamination and growth of microorganisms. Basil (*Ocimum basilicum* L.) EO (OBEO) was analyzed as a possible treatment to eliminate synthetic preservatives and to reduce storage losses. OBEO proved highly active against *Colletotrichum gleosporioides* that causes diseases in mangoes [10]. In the case of grapes stored at 0 °C, OBEO inhibited the development of fungal decay for up to 60 days, but also limited weight loss and browning, without significantly affecting the taste of the fruit [11]. Other studies indicated the possibility of extending the shelf life of grapes by dipping them in a 20% eugenol solution and packing under a modified atmosphere [12]. Herath and Abeywickrama [13] showed that OBEO and eugenol significantly reduced the prevalence of banana crown rot fungi, *Colletotrichum musae* and *Fusarium proliferatum*. Another study showed that linalool vapors limited the rotting of peach fruits caused by *Rhizopus stolonifer*, *Sclerotinia sclerotiorum* and *Mucor* sp. [14]. The combination of linalool and eugenol in the proportions found naturally in OBEO resulted in a significant increase in antimicrobial activity.

Moghaddam et al. [15] described the antibacterial activity of OBEO, in particular of methyl chavicol, against this important phytopathogenic bacteria. The excessive use of synthetic plant protection products, such as fungicides, insecticides or pesticides, which are also toxic to other organisms, causes substantial environmental and toxicological problems. Therefore, it is important to replace the synthetic plant protection products with other, more environmentally friendly alternatives [5,16]. The use of biological plant protection is safer also because of its environmental impacts. Therefore, in recent years, there has been an increasing interest in the use of natural and plant-based pest control products such as EO or their derivatives. The most frequently mentioned arguments in their support, in contrast to the synthetic products are, much lower toxicity to mammals, but also rapid degradation in the environment [17,18]. Literature reports indicate the wide antimicrobial or insecticidal activity of EOs, which allows their application for crop plant protection [19,20,21].

Considering the above, the aim of this study was to characterize the chemical composition of the essential oil of *Ocimum basilicum* (OBEO) as well as its antioxidant, anti-insect and antimicrobial activities in an in vitro and in situ model of fruit and vegetables.

## 2. Results

### 2.1. Chemical Composition of OBEO

The chemical composition of OBEO was determined by GC/MS analyses. The identified volatile compounds and their relative content (percentages) are listed in Table 1. In total, 28 compounds were identified in OBEO; those accounted for 99.8% of the total volatiles. The obtained data revealed that the main component of the OBEO was methyl chavicol (88.6%), followed by 1,8-cineole (4.2%) and *α*-trans-bergamotene (1.7%).

### 2.2. Antioxidant Activity of OBEO

The antioxidant activity of OBEO assessed by the DPPH method was determined at 77.3 ± 5.0% of inhibition that corresponds to 434.21 ± 28.9 µg of TEAC/mL (trolox equivalent antioxidant capacity).

### 2.3. In Vitro Antimicrobial Activity of OBEO

In the analysis of OBEO antimicrobial activity, G^−^ bacteria, G^+^ bacteria, and yeasts were used. The OBEO showed a varying activity against the growth of the microorganisms used in the disc diffusion tests (Table 2). The G^−^ bacteria with inhibition zones 14.33 ± 0.58 mm (*A. chrococcum*) and 15.33 ± 0.55 mm (*S. marcescens*) were more sensitive to OBEO than the G^+^ bacteria with inhibition zones of 9.67 ± 0.58 mm (*M. luteus*) and 11.33 ± 0.58 mm (*P. megatherinum*). The yeasts were the least sensitive to OBEO, with the inhibition zones of 5.33 ± 0.58 mm (*C. glabrata*) and 7.33 ± 0.58 mm (*C. tropicalis*), respectively. 

The minimal inhibition concentration (MIC) of OBEO is shown Table 3. The MIC test showed the strongest inhibition activity and the lowest MIC 50 (3.21 μL/mL) and MIC 90 (5.36 μL/mL) values for *A. chrococcum* and *M. luteus*. Conversely, the highest MIC 50 (6.15 μL/mL) and MIC 90 (8.25 µL/mL) values were recorded for *C. glabrata*.

### 2.4. In Situ Antimicrobial Activity of OBEO

The results of the antibacterial activity of the vapor phase of OBEO on apples are summarized in Table 4. Intensity of the bacterial inhibition by OBEO increased with the increasing concentration of OBEO in assays across all bacteria tested.

The results of the anti-yeasts activity of the vapor phase of OBEO on apples are summarized in Table 5. Intensity of the yest growth inhibition by OBEO increased with the increasing concentration of OBEO in assays across all bacteria tested. In agreement with the MIC results, *C. glabrata* was more sensitive to the OBEO than *C. tropicalis*, in particular at the higher tested concentrations.

The results of the antibacterial activity of the vapor phase of OBEO on pears are summarized in Table 6. Intensity of the bacterial inhibition by OBEO increased with the increasing concentration of OBEO in assays across all bacteria tested. The best antibacterial activity was found against *S. marcescens* at 500 μL/L concentration. In contrast to the similar assays in apples, *P. megatherium* was inhibited by OBEO the lowest in the pear assays.

The results of the anti-yeast activity of the vapor phase of OBEO on pears are summarized in Table 7. Similar to the MIC results, *C. glabrata* was more sensitive to the OBEO than *C. tropicalis*, in particular at the highest tested concentration.

The results of the antibacterial activity of the gas phase of OBEO on potatoes are summarized in Table 8. Similar to the other inoculated plants, there was an increasing trend of bacterial inhibition with the increasing amounts of the applied OBEO. The best antibacterial activity of OBEO on potatoes was found against *M. luteus* at 500 μL/L concentration.

The results of the anti-yeast activity of the gas phase of OBEO on potatoes are summarized in Table 9. Again, application of higher amounts of OBEO resulted in higher levels of yeast growth inhibition, and *C. glabrata* was inhibited marginally higher than *C. tropicalis*. 

The results of the antibacterial activity of the vapor phase of OBEO on kohlrabies are summarized in Table 10. In agreement with other tested plants, increasing the amounts of applied OBEO resulted in increased bacterial inhibition. The best antibacterial activity was found against *S. marcescens* at 500 μL/L concentration reaching 98.95%.

The results of the anti-yeast activity of the gas phase of OBEO on kohlrabies are summarized in Table 11. Congruently with other tested plants, the inhibitory activity was higher at the higher OBEO concentration applied. In contrast with other tested plat species, *C. tropicalis* showed marginally higher sensitivity to OBEO than *C. glabrata*. 

### 2.5. Anti-Insect Activity of OBEO

All tested concentrations of the OBEO vapor phase showed insecticidal activity against *Pyrrhocoris apterus* (Table 12). The highest insecticidal activity was observed at the concentration of 100%, causing the death of 80% of individuals. About a quarter of the OBEO concentration resulted in roughly half the mortality, whereas 6.25% of OBEO applied proved indistinguishable from the control treatment.

## 3. Discussion

Research on basil oil has showed that it has antimicrobial and insecticidal activity as well as the high antioxidant activity, antiprotozoal activity and anticancer effects, and therefore, it has been used as the flavoring agent in gastronomy, for food preservation, safety and production. There were four chemotypes of basil identified [22]; the European chemotype contained linalool as the main component (Italy, France, Bulgaria, Egypt and South Africa), the tropical chemotype with the major component of methyl cinnamate (India, Pakistan, Guatemala), the Reunion chemotype with methyl chavicol (Madagascar, Thailand, Vietman) [23,24] and the North Africa and Russia chemotype with eugenol [23]. In other studies, the major component of basil essential oil was estragole (methyl chavicol) [25,26,27,28,29,30].

We observed a high antioxidant activity of OBEO in our study. Other previous research documented that *Ocimum* sp. contained a high content of antioxidant compounds which could contribute to increasing the OBEO antiradical activity [31,32] and positively affecting human health [33]. Basil is characterized by a strong antioxidant capacity so its use can help in prevention of heart diseases, and help decrease the inflammation and the incidence of cancers and diabetes [34]. A strong correlation was found between phenolic compounds and antioxidant capacity of medicinal plants [35,36]. Extracts of *Ocimum basilicum* consist in a substantial part of phenolic acids and thus show high antioxidant activity. The antioxidant activity of *O. basilicum* var. *cinammom* caused an antioxidant inhibition of 87.9% at a concentration of 10 ppm, an inhibition of 91.4% at 30 ppm and 92.1% up to 50 ppm. EOs of *O. basilicum* var. *album* showed a 61.8% antioxidant inhibition at 10 ppm and 90% inhibition at 30 ppm. Inhibition remained constant at 50 ppm [37].

We tested the effects of OBEO application under in vitro conditions against G^−^, G^+^ and yeasts in our study. Similar to our results, there is evidence of an antibacterial activity of EOs from various *Ocimum* species in vitro against *Staphylococcus aureus*, *Salmonella enteritidis*, *Escherichia coli*, *Proteus vulgaris*, *Bacillus subtilis*, *Salmonella typhi*, *Shigella sonnei*, *Shigella boydii*, *Pseudomonas aeruginosa* and *Salmonella paratyphi* [38,39,40]. OBEO activity was also tested against pathogenic fungi such as: *Aspergillus niger*, *Aspergillus fumigatus*, *Penicillium italicum* and *Rhizopus stolonifera* by using a disc diffusion method, and by determination of minimum inhibitory concentration. High antifungal activity found in that study highlighted the potential of *Ocimum* species to be used as a preservative in food and pharmaceutical industries [41]. We found the highest antimicrobial activity using the disc diffusion method against *A. chrococcum* and these results were confirmed by the agar diffusion assays. Our results confirm many other studied of the antibacterial properties of *O. basilicum*. Chenni et al. [42] reported that the EOs extracted from *O. basilicum* leaves using steam distillation displayed suitable antibacterial activity on G^−^ and G^+^ bacteria including *E. coli*, *P. aeruginosa*, *Bacillus cereus* and *S. aureus*. The results of another study reported the minimum inhibitory concentration (MIC) of *O. basilicum* extracts against *B. cereus* (36–18 μg/mL), *S. aureus* (18 μg/mL) and *E. coli* (9–18 μg/mL) [43]. The antibacterial activity of *O. basilicum* EOs has also been studied in Vietnam and the results have been very promising. Vu et al. [44] used in their study the Thai basil grown and harvested in Tien Giang and found out that the EO can inhibit some bacteria that cause intestinal diseases, such as *E. coli*, *Salmonella typhimurium*, *Shigella*, *B. cereus*, methicillin-resistant *Staphylococcus aureus* (MRSA), but not *P. aeruginosa*. The anti-yeast activity of the OBEO was previously tested on *Candida albicans* [45].

The postharvest research attracted a substantial scientific and industrial interest; it is focused on phytonutrients that primarily increase the nutritional value of fruit and vegetables in addition to the reduction of the losses during storage [46]. Basil EOs showed high activities against postharvest diseases caused by *Fusarium* fungi [47]. Previous research showed a significant inhibition of pathogenic molds *B. cinerea* and *Mucor piriformis* by sweet basil EOs documented in in vitro tests by EOs incorporation in media or in vapor [11]. The basil EOs showed an inhibitory effect against G^−^ bacteria *Shigella sonnei* and *Shigella flexneri* on lettuce. In another study, washing fresh lettuce with 0.1 or 1% (*v*/*v*) of estragole chemotype basil EOs showed similar effectiveness to natural microbial flora as treatment with 125 ppm chlorine [48].

The plant pathogenic fungus *Botrytis fabae* and the rust fungus *Uromyces fabae* were controlled in vivo using basil EOs, estragole and linalool; these treatments significantly reduced the infection of broad bean leaves [49]. Basil EOs at 0.08% mL/L showed some efficiency against the survival of *E. coli* and *Salmonella typhimurium* inoculated into fresh-cut lettuce and purslane samples during refrigerated storage [50]. The results of our study showed a strong activity of OBEO against G^−^, G^+^ and yeast under in situ study on apples, pears, potatoes and kohlrabies. The highest antimicrobial effect of OBEO was found against *Serratia marcescens* on fruits and vegetables samples. The physiological quality of indicators such as fruit firmness depends on the EO dose applied. Crop firmness is one of the most important indicators used for the quality assessment of fruits. Our previous study used a vapor phase at 62.5 to 500 µL/L [14]. The minimum fungicidal dose in vapor phase was 300 µL/L for basil EO on lemon fruits [51]. The fungicidal activity of basil EO in vapor phase ranged between 30 to 400 µL/L [14]. A number of samples assayed by vapor phase in other studies were tested at relatively high concentrations (e.g., between 800 and 1600 mg/L); such high concentrations are only suitable for a qualitative evaluation, and can lead to a misinterpretation for the practical application of the obtained results [52].

The loss of fruit firmness during storage points out to water losses and metabolic changes in fruits [53]. Use of various polysaccharide, protein or carboxymethyl cellulose (CMC) coating techniques or edible crop coatings with chitosan bilayers did not help form an effective water-vapor barriers [54]. Therefore, the EO vapor-phase treatment can be more efficient during the post-harvest treatment of fruits. The research on the application of cinnamon and eucalyptus EOs vapor treatments at 50 ppm showed positive effects on maintaining the firmness in tomato and cherry tomatoes [53]. Firmness, texture and resistance to microbial infection in shiitake mushrooms was improved after fumigation with cloves, thymus and cinnamaldehyde EOs vapors for 20 days at 4 °C storage [55]. The tested samples of mushrooms were better-preserved in comparison to the control samples, but there was also a higher concentration of EOs with an undesirable effect observed. Due to their high volatility, EOs belong to the natural method of crop preservation with a high potential of use mainly in the organically grown fruit and vegetables. Furthermore, EOs utilized in the vapor-phase could be effective against fungal pathogens [51].

We observed a strong anti-insect activity of OBEO in higher concentration against *P. apterus*. Basil plants are attractive for many insects, including butterflies, bees, fruit flies and many others [56]. The addition of basil EOs on sticky traps increased approximately three and fivefold the number of trapped pea leafminer (*Liriomyza huidobrensis*) and greenhouse whitefly (*Trialeurodes vaporariorum*), respectively [57]. Sweet basil EOs showed antifeedant and insecticidal activities against the cowpea aphid *Aphis craccivora*. The successful adoption of basil EOs in the protection of food commodities promises an eco-friendly option compatible with the international biosafety regulations and the demands of organic certifications.

In conclusion, basil EOs demonstrate the desirable antimicrobial and anti-insect effects in both in vitro and in situ studies. OBOE shows a great potential in organic product cultivation and food preservation [58]. However, a number of previous studies were done under limiting conditions in laboratories so an in-field analysis of the OBOE efficacy is needed [59].

Currently, the number of commercially available biocontrol products using EOs as active components is limited due to their selective effectiveness [60]. The EOs efficacies vary among the pathogen species, sometimes even among strains of the same pathogen species. The EOs may display anti-yeast effects and phytotoxic effects at the same time [61]. It is still difficult to find a proper application concentration for each crop at each developmental stage to optimally suppress the pathogens without damaging the crop plants.

As consumers become increasingly more aware of the human and environmental health, they are willing to pay a higher price for the organic commodities. Therefore, theoretical and practical studies of EOs should continue and more focus is needed on the development, extension and market introduction of newly formulated EO products [62].

## 4. Materials and Methods

### 4.1. Sample

The test essential oil *Ocimum basilicum* (OBEO) was purchased from company Hanus s.r.o. (Nitra, Slovakia) prepared by steam distillation of a fresh flowering plants from Vietnam. EO was stored in a refrigerator (4 °C) protected from light, in glass vessels.

### 4.2. Chemical Composition of OBEO

GC/MS analyses of OBEO were performed using an Agilent 6890N gas chromatograph (Agilent Technologies, Santa Clara, CA, USA) coupled to quadrupole mass spectrometer 5975B (Agilent Technologies, Santa Clara, CA, USA). A HP-5MS capillary column (30 m × 0.25 mm × 0.25 mm) was used. Temperature program was: 60 °C to 150 °C (increasing rate 3 °C/min) and 150 °C to 280 °C (increasing rate 5 °C/min). The total run time was 60 min. Helium 5.0 was used as the carrier gas with flow rate of 1 mL/min. The injection volume was 1 mL (essential oil sample was diluted in pentane), while the split/splitless injector temperature was set at 280 °C. With split ratio at 40.8:1 investigated samples were injected in the split mode. Electron-impact mass spectrometric data (EI-MS; 70 eV) were acquired in scan mode over the m/z range 35–550. MS ion source and MS quadrupole temperatures were 230 °C and 150 °C, respectively. Acquisition of data started after solvent delay time of 3 min. GC-FID analyses were performed on Agilent 6890N gas chromatograph coupled to FID detector. Column (HP-5MS) and chromatographic conditions were same as for GC-MS. FID detector temperature was set at 300 °C.

The individual volatile constituents of the injected essential oil sample were identified based of their retention indices and comparison with reference spectra (Wiley and NIST databases). The retention indices were experimentally determined using the standard method which included retention times of n-alkanes (C6–C34), injected under the same chromatographic conditions. The percentages of the identified compounds (amounts higher than 0.1%) were derived from their GC peak areas [63].

### 4.3. Antioxidant Activity of OBEO

The antioxidant activity was determined spectrophotometrically by the DPPH method [64]. The percentage of inhibition was calculated as (A0−AA)/A0 × 100, where A0 was the absorbance of blank measurement, and AA was absorbance of sample. The antioxidant activity was expressed as antioxidant activity of Trolox related to 1 mL of sample (µg of TEAC/mL). Measurements were done in triplicate.

### 4.4. Microorganisms Tested

Gram-negative (G^−^) bacteria (*Azotobacter chrococcum* CCM 1912, *Serratia marcescens* CCM 8587), Gram-positive (G^+^) bacteria (*Bacillus* (isolated from 2020 *Priestia megatherium* CCM 2007, *Micrococcus luteus* CCM 732) and yeasts (*Candida glabrata* CCM 8270, *C. tropicalis* CCM 8223,) were obtained from the Czech Collection of Microorganisms (CCM; Brno, Czech Republic).

### 4.5. In Vitro Antimicrobial Activity of OBEO

The disc diffusion method was used to assess the antimicrobial activity of OBEO. The inoculum of bacteria was grown at 37 °C for 24 h on tryptone soy agar (Oxoid, Basingstoke, UK), whereas the yeast inoculum was grown at 25 °C on Sabouraud dextrose agar (Oxoid, Basingstoke, UK). The obtained inoculum was then adjusted to the optical density of a 0.5 McFarland standard (1.5 × 10^8^ CFU/mL) then 100 µL was placed in a Petri dish (PD) with Mueller Hinton medium (Oxoid, Basingstoke, UK) for bacteria and Sabouraud dextrose agar for yeast. Then, sterile disks with a diameter of 6 mm were placed on the solidified medium, onto which 10 µL of OBEO was applied. The PD prepared in this way was incubated in the above-mentioned conditions for the next 24 h. Antibiotics were used as a positive control for bacteria: cefoxitin for Gram-negative and gentamicin for Gram-positive, and for yeasts, an antifungal agent was used—fluconazole. All substances were obtained from Oxoid (Basingstoke, UK). As a negative control, discs with 10 µL of 0.1% DMSO applied on them (Centralchem, Bratislava, SK) were used. After incubation, the zones of growth inhibition were determined: above 10 mm antimicrobial activity was determined as very strong, above 5 mm as mild and above 1 mm as weak [63].

### 4.6. Minimum Inhibitory Concentration (MIC)

The MICs of bacteria and yeasts were determined using the agar microdilution method. The inoculum was cultured for 24 h in Mueller Hinton Broth (MHB, Oxoid, Basingstoke, UK) at 37 °C for bacteria and Sabouraud Dextrose Broth (SDB, Oxoid, Basingstoke, UK) at 25 °C for yeast. Then, 100 µL of nutrient medium and 50 µL of inoculum with an optical density of 0.5 McFarland standard were applied to a 96-well microtiter plate. Subsequently, OBEO was prepared by serial dilution to a concentration range of 400 µL/mL to 0.2 µL/mL in MHB/SDB and mixed thoroughly with bacterial inoculum in the wells. The prepared 96-well microtiter plates were measured at 570 nm with a Glomax spectrophotometer (Promega Inc., Madison, WI, USA) at 0 h. Subsequently, the bacterial samples were incubated at 37 °C for 24 h. Yeast samples were incubated at 25 °C for 24 h and measured again. MHB/SDB with essential oil was used as a negative control, and MHB/SDB with inoculum was used as a positive control for maximal growth. The analysis was performed in triplicate [65,66,67].

### 4.7. In Situ Antimicrobial Activity of OBEO

The evaluation of the antimicrobial activity of OBCE on fruit and vegetable models (apples, pears, potatoes and kohlrabi) was performed according to the described by Borotová et al. [68] procedure against four bacteria and two yeasts. Briefly, 5 mm thick slices of fruit and vegetables were placed on solidified Mueller Hinton agar for PD (∅ = 60 mm) and a microbial inoculum (0.5 McFarland) was applied. Then, diluted OBEO (100 µL) in ethyl acetate at 4 dilution levels (500, 250, 125 and 62.5 µL/L) was applied to a disc of sterile filter paper. The hermetically sealed PD was incubated for 7 days at the temperature appropriate for the analyzed microorganisms. An equivalent volume of ethyl acetate was used as a negative control. The percentage of inhibitory activity was calculated in ImageJ by stereological method. Bulk density was calculated according to the formula Vv = P/p × 100 where P is stereological lattice of the colonies and p is the substrate. Growth inhibition was expressed as GI = [(C − T)/C] × 100, where C was the growth density of control group and T was the growth density in the group contained LCEO [69].

### 4.8. Anti-Insect Activity of OBEO

The insecticidal activity of OBEO was tested using *Pyrrhocoris apterus* in accordance with the previously described method [69]. Briefly, concentrations of 100%, 50%, 25%, 12.5% and 6.25% OBEO were tested, and a 0.1% polysorbate was used as the negative control. A total of 30 individuals of *P. apterus* were placed in PD with vents. One hundred µL of the appropriate concentration of OBEO was placed on a filter paper wheel, placed in the cover PD and the plates sealed with parafilm. After 24 h of exposure at room temperature, the numbers of living and dead *P. apterus* were assessed, and then the percentage of insecticidal activity was calculated. Analyses were carried out in triplicate.

### 4.9. Statistical Data Evaluation

One-way analysis of variance (ANOVA) was performed using Prism 8.0.1 (GraphPad Software, San Diego, CA, USA) followed by Tukey’s test at *α* = 0.05. SAS^®^ software version 8 was used for data processing. The results of the MIC50 value (concentrations that caused 50% of microbial growth) and MIC90 (90% inhibition of bacterial growth) were determined by logit analysis.

## 5. Conclusions

Our study details the chemical composition, antioxidant, antimicrobial (in vitro, in situ) and anti-insect properties of commercial OBEO. The results obtained in this study strongly underscore that basil EOs could be used as the candidate for the development of natural antibiotics and disinfectants to control microbes pathogenic to crop fruit and vegetables. Thus, the basil EOs could be used as a naturally sourced food preservative to reduce and substitute or avoid the use of chemical preservatives. The results of our study and others encourage and prompt larger-scale, in-field research on basil EOs application, in particular for organic farming.

## Figures and Tables

**Table 1 plants-11-01030-t001:** Chemical composition of essential oil from *Ocimum basilicum*.

No	RI ^a^	Compound ^b^	% ^c^
1	938	*α*-pinene	1.0
2	948	camphene	tr
3	977	sabinene	tr
4	980	*β*-pinene	0.2
5	1023	*p*-cimene	0.7
6	1028	*α*-limonene	0.4
7	1033	1,8-cineole	4.2
8	1047	(*E*)-*β*-ocimene	0.4
9	1060	*γ*-terpinene	0.4
10	1085	fenchone	tr
11	1088	*α*-terpinolene	1.2
12	1148	camphor	0.3
13	1195	methyl chavicol	88.6
14	1223	endo-fenchyl acetate	tr
15	1235	exo-fenchyl acetate	tr
16	1286	bornyl acetate	tr
17	1360	eugenol	0.3
18	1388	*β*-elemene	tr
19	1406	methyl eugenol	0.2
20	1422	(*E*)-caryophyllene	tr
21	1437	*α*-trans-bergamotene	1.7
22	1443	(*Z*)-*β*-farnesene	tr
26	1485	*α*-amorphene	0.2
27	1568	(*E*)-*p*-methoxy-cinnamaldehyde	tr
28	1583	caryophyllene oxide	tr
	total		99.8

^a^ Values of retention indices on HP-5MS column; ^b^ Identified compounds; ^c^ tr–compounds identified in amounts less than 0.1%.

**Table 2 plants-11-01030-t002:** In vitro analysis of the antimicrobial activity of the OBEO with disc diffusion method.

Microorganisms	Inhibition Zone (mm)	Activity of Basil EO	ATB (mm)
Gram-negative bacteria			
*Azotobacter chrococcum*	15.33 ± 0.58	***	26.33 ± 1.53
*Serratia marcescens*	14.33 ± 0.58	***	30.33 ± 0.58
Gram-positive bacteria			
*Priestia megatherium*	11.33 ± 0.58	***	30.33 ± 0.58
*Micrococcus luteus*	9.67 ± 0.58	**	27.33 ± 0.58
Yeasts			
*Candida glabrata*	5.33 ± 0.58	**	28.33 ± 0.58
*Candida tropicalis*	7.33 ± 0.58	**	29.33 ± 1.15

Means ± standard deviation (*n* = 3). ** Moderate inhibitory activity (zone > 10 mm). *** Very strong inhibitory activity (zone > 15 mm). ATB—antibiotics, positive control (Cefoxitin for G^−^, Gentamicin for G^+^, Fluconazole for yeast).

**Table 3 plants-11-01030-t003:** Minimal inhibition concentration of OBEO.

Microorganisms	MIC50 (µL/mL)	MIC90 (µL/mL)
Gram-negative bacteria		
*Azotobacter chrococcum*	3.21	5.36
*Serratia marcescens*	3.46	6.23
Gram-positive bacteria		
*Priestia megatherium*	3.25	5.41
*Micrococcus luteus*	3.21	5.36
Yeasts		
*Candida glabrata*	6.15	8.25
*Candida tropicalis*	3.56	5.67

**Table 4 plants-11-01030-t004:** In situ analysis of the antibacterial activity of the vapor phase of OBEO on apples.

Basil EO (µL/L)	Bacterial Growth Inhibition [%] Apples			
	*Azotobacter chrococcum*	*Priestia megatherium*	*Serratia marcescens*	*Micrococcus luteus*
62.5	12.31 ± 1.97 ^a^	13.07 ± 2.44 ^a^	14.87 ± 2.29 ^a^	7.79 ± 2.55 ^a^
125	24.69 ± 2.45 ^b,a^	17.78 ± 2.61 ^b^	28.71 ± 3.46 ^b^	14.38 ± 2.89 ^b^
250	53.19 ± 2.57 ^c,a,b^	35.91 ± 1.51 ^c,a,b^	72.09 ± 3.07 ^c,a,^	26.00 ± 2.56 ^c,a,b^
500	92.35 ± 3.56 ^d,a,b,c^	80.81 ± 2.11 ^d,a,b,c^	99.41 ± 1.01 ^d,a,b^	74.13 ± 3.13 ^d,a,b,c^

Means ± standard deviation (*n* = 3). Individual letters (a–d) in upper case indicate the statistical difference at *p* ≤ 0.05.

**Table 5 plants-11-01030-t005:** In situ analysis of the anti-yeast activity of the vapor phase of OBEO on apples.

Basil EO (µL/L)	Mycelial Growth Inhibition [%] Apples	
	*Candida glabrata*	*Candida tropicalis*
62.5	13.56 ± 2.94 ^a^	13.61 ± 3.82 ^a^
125	33.30 ± 2.48 ^b,a^	16.58 ± 2.70 ^b^
250	45.72 ± 3.40 ^c,a,b^	28.88 ± 2.88 ^c,a,b^
500	74.42 ± 3.64 ^d,a,b,c^	27.56 ± 4.09 ^d,a,b,c^

Means ± standard deviation (*n* = 3). Individual letters (a–d) in upper case indicate the statistical difference at *p* ≤ 0.05.

**Table 6 plants-11-01030-t006:** In situ analysis of the antibacterial activity of the vapor phase of OBEO on pears.

Basil EO (µL/L)	Bacterial Growth Inhibition [%] Pears			
	*Azotobacter chrococcum*	*Priestia megatherium*	*Serratia marcescens*	*Micrococcus luteus*
62.5	6.39 ± 1.66 ^a^	1.85 ± 1.47 ^a^	2.51 ± 0.99 ^a^	10.17 ± 2.20 ^a^
125	13.18 ± 1.16 ^b,a^	3.18 ± 1.29 ^b^	7.54 ± 2.87 ^b^	12.65 ± 2.50 ^b^
250	35.34 ± 3.93 ^c,a,b^	5.16 ± 2.39 ^c^	16.09 ± 3.98 ^c,a,b^	28.61 ± 3.11 ^c,a,b^
500	73.11 ± 2.02 ^d,a,b,c^	16.02 ± 2.54 ^d,a,b,c^	82.38 ± 3.72 ^d,a,b,c^	63.40 ± 2.78 ^d,a,b,c^

Means ± standard deviation (*n* = 3). Individual letters (a–d) in upper case indicate the statistical difference at *p* ≤ 0.05.

**Table 7 plants-11-01030-t007:** In situ analysis of the anti-yeast activity of the vapor phase of OBEO on pears.

Basil EO (µL/L)	Mycelial Growth Inhibition [%] Pear	
	*Candida glabrata*	*Candida tropicalis*
62.5	4.61 ± 0.74 ^a^	4.79 ± 2.13 ^a^
125	12.35 ± 2.05 ^b^	17.38 ± 1.89 ^b,a^
250	30.37 ± 5.00 ^c,a,b^	23.19 ± 1.51 ^c,a,b^
500	75.38 ± 3.89 ^d,a,b,c^	45.25 ± 2.70 ^d,a,b,c^

Means ± standard deviation (*n* = 3). Individual letters (a–d) in upper case indicate the statistical difference at *p* ≤ 0.05.

**Table 8 plants-11-01030-t008:** In situ analysis of the antibacterial activity of the vapor phase of OBEO on potatoes.

Basil EO (µL/L)	Bacterial Growth Inhibition [%] Potatoes			
	*Azotobacter chrococcum*	*Priestia megatherium*	*Serratia marcescens*	*Micrococcus luteus*
62.5	17.27 ± 2.01 ^a^	4.61 ± 0.74 ^a^	12.12 ± 2.12 ^a^	15.47 ± 3.03 ^a^
125	25.38 ± 2.82 ^b^	13.30 ± 2.01 ^b,a^	19.82 ± 1.43 ^b,a^	4.82 ± 2.16 ^b,a^
250	46.52 ± 2.70 ^c,a,b^	31.38 ± 2.86 ^c,a,b^	6.49 ± 0.69 ^c,b^	46.70 ± 1.29 ^c,a,b^
500	67.02 ± 4.55 ^d,a,b,c^	72.04 ± 3.13 ^d,a,b,c^	39.89 ± 4.13 ^d,a,b,c^	79.74 ± 1.57 ^d,a,b,c^

Means ± standard deviation (*n* = 3). Individual letters (a–d) in upper case indicate the statistical difference at *p* ≤ 0.05.

**Table 9 plants-11-01030-t009:** In situ analysis of the anti-yeast activity of the vapor phase of OBEO on potatoes.

Basil EO (µL/L)	Mycelial Growth Inhibition [%] Potatoes	
	*Candida glabrata*	*Candida tropicalis*
62.5	7.74 ± 1.78 ^a^	15.94 ± 2.43 ^a^
125	21.21 ± 2.60 ^b,a^	26.65 ± 3.13 ^b,a^
250	46.82 ± 3.26 ^c,a,b^	44.07 ± 2.16 ^c,a,b^
500	87.31 ± 2.71 ^d,a,b,c^	78.04 ± 3.95 ^d,a,b,c^

Means ± standard deviation (*n* = 3). Individual letters (a–d) in upper case indicate the statistical difference at *p* ≤ 0.05.

**Table 10 plants-11-01030-t010:** In situ analysis of the antibacterial activity of the vapor phase of OBEO on kohlrabies.

Basil EO (µL/L)	Bacterial Growth Inhibition [%] Kohlrabies			
	*Azotobacter chrococcum*	*Priestia megatherium*	*Serratia marcescens*	*Micrococcus luteus*
62.5	10.93 ± 1.28 ^a^	27.35 ± 1.60 ^a^	12.39 ± 2.19 ^a^	12.68 ± 2.62 ^a^
125	23.46 ± 2.57 ^b,a^	35.89 ± 2.41 ^b,a^	25.61 ± 3.10 ^b,a^	25.30 ± 3.17 ^b,a^
250	55.15 ± 2.47 ^c,a,b^	43.81 ± 2.56 ^c,a,b^	55.04 ± 3.12 ^c,a,b^	7.77 ± 2.03 ^c,b^
500	75.11 ± 2.90 ^d,a,b,c^	55.45 ± 4.59 ^d,a,b,c^	98.95 ± 0.97 ^d,a,b,c^	6.07 ± 2.17 ^d,b^

Means ± standard deviation (*n* = 3). Individual letters (a–d) in upper case indicate the statistical difference at *p* ≤ 0.05.

**Table 11 plants-11-01030-t011:** In situ analysis of the anti-yeast activity of the vapor phase of OBEO on kohlrabies.

Basil EO (µL/L)	Mycelial Growth Inhibition [%] Kohlrabies	
	*Candida glabrata*	*Candida tropicalis*
62.5	6.44 ± 1.84 ^a^	8.50 ± 1.11 ^a^
125	13.27 ± 2.25 ^b,a^	18.05 ± 4.50 ^b^
250	31.59 ± 1.48 ^c,a,b^	38.55 ± 5.71 ^c,a,b^
500	67.70 ± 2.13 ^d,a,b,c^	74.91 ± 0.66 ^d,a,b,c^

Means ± standard deviation (*n* = 3). Individual letters (a–d) in upper case indicate the statistical difference at *p* ≤ 0.05.

**Table 12 plants-11-01030-t012:** Anti-Insect activity of OBEO.

Concentration [%]	Number of Living Individuals	Number of Dead Individuals	Insecticidal Activity [%]
100	4	26	80
50	9	21	70
25	15	15	50
12.5	27	3	10
6.25	30	0	0
Control group	30	0	0

## Data Availability

Data is contained within the article.

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
