# Peer review of "Assessment of Ocimum basilicum Essential Oil Anti-Insect Activity and Antimicrobial Protection in Fruit and Vegetable Quality"

_plants, 2022, doi:10.3390/plants11081030_

Round 1

Reviewer 1 Report

The aim of the research is interesting. The introduction is detailed, and the discussion is well organized, highlighting also the drawbacks of the application of EOs as biocontrol products. 

On the contrary, the methods are poor, the experiments have to be described in detail even though the methods are reported elsewhere. There are several issues that are not clear. 

  •  Section 4.1: explain the steam distillation methodology.
  • Section 4.2: provide the chromatographic conditions.
  • section 4.3: Provide details of the assays and explain how the percent inhibition was calculated.
  • Line 346 and 364: how did you dilute the EO? Did you add the diluting agent to the controls?
  • Why did you use polysorbate as negative control?
  • it is not clear how the authors calculated the percentages of inhibition in the in situ assays.
  • Moreover, you have to provide the results of the negative and positive control for all the assays.

Minor observations: 

  • Line 104: specify what “TEAC” is
  • Line 132:  correct the word “yest”

Author Response

Reviewer #1

The aim of the research is interesting. The introduction is detailed, and the discussion is well organized, highlighting also the drawbacks of the application of EOs as biocontrol products. 

Thank you very much for the positive evaluation and for your time.

Point 1: On the contrary, the methods are poor, the experiments have to be described in detail even though the methods are reported elsewhere. There are several issues that are not clear. Section 4.1: explain the steam distillation methodology.

Response: Essential oil was commercially produced.  Steam distillation is carried out by passing dry steam through the plant material whereby the steam volatile compounds are volatilized, condensed and collected in receivers. Steam distillation has been in use for essential oil extraction for many years. The text was added to manuscript.

Point 2: Section 4.2: provide the chromatographic conditions.

Response: It was added.

Point 3:  Section 4.3: Provide details of the assays and explain how the percent inhibition was calculated.

Response: It was added.

Point 4: Line 346 and 364: how did you dilute the EO? Did you add the diluting agent to the controls?

Response: It was changed.

Point 5: Why did you use polysorbate as negative control?

Response: 0.1% polysorbate was used for dilution also in situ antimicrobial activity. Polysorbate is not toxic for insects.

Point 6: It is not clear how the authors calculated the percentages of inhibition in the in situ assays.

Response: Calculation of percentage was added to materials and methods.

Point 7: Moreover, you have to provide the results of the negative and positive control for all the assays.

Response: Yes, the negative and positive control for all assays were used and it is now described in materials and methods.

Point 8: Minor observations: 

  • Line 104: specify what “TEAC” is  

Response: TEAC is trolox equivalent antioxidant capacity

  • Line 132:  correct the word “yest”

Response: It was corrected.

Reviewer 2 Report

This study has aimed to characterize the chemical composition, antioxidant, and antimicrobial activities of the essential oil of Ocimum basilicum. Generally, it is an interesting study that deserves to be published. There are some suggestions:

Many of the sentences don’t have coherence. For example, the first two sentences under ‘introduction’ don’t have coherence.

I would suggest adding the disc diffusion method photos in supplementary materials

Please add the distillation yield.

Please substantiate more the methods section 4.2 to allow replication of results.

Please add more information regarding Pyrrhocoris apterus and why you choose this insect for testing the oils.

Author Response

Reviewer #2

This study has aimed to characterize the chemical composition, antioxidant, and antimicrobial activities of the essential oil of Ocimum basilicum. Generally, it is an interesting study that deserves to be published. There are some suggestions:

Thank you very much for the positive evaluation and for your time.

Point 1: Many of the sentences don’t have coherence. For example, the first two sentences under ‘introduction’ don’t have coherence.

Response: It was changed.

Point 2: I would suggest adding the disc diffusion method photos in supplementary materials.

Response: Sorry reviewer, we don’t have these photos.

Point 3: Please add the distillation yield.

Response: Essential oil was commercially produced.  Steam distillation is carried out by passing dry steam through the plant material whereby the steam volatile compounds are volatilized, condensed and collected in receivers. Steam distillation has been in use for essential oil extraction for many years. The text was added to manuscript.

Point 4: Please substantiate more the methods section 4.2 to allow replication of results.

Response: It was added to material and methods.

Point 5: Please add more information regarding Pyrrhocoris apterus and why you choose this insect for testing the oils.

Response: P. apterus was a model organism used. It was insects which were in disposition in time when we tested EOs. In the future we will continue testing insects dangerous for fruits and vegetables.

Reviewer 3 Report

The paper addresses a topical issue, namely the use of natural and plant-based pest control products.

The research is conducted correctly in terms of the sequence of steps.

Actual results are clearly presented and support the conclusions. The bibliography is current and exhaustive, correctly written.

I have only a small critical remark: you must review the text R263-264 (typo error).

Author Response

Reviewer #3

The paper addresses a topical issue, namely the use of natural and plant-based pest control products. The research is conducted correctly in terms of the sequence of steps. Actual results are clearly presented and support the conclusions. The bibliography is current and exhaustive, correctly written.

Thank you very much for the positive evaluation and for your time.

Point 1: I have only a small critical remark: you must review the text R263-264 (typo error).

Response: It was corrected.

Reviewer 4 Report

It is an interesting and up-to-date study of the bioactivity of basil essential oil. The strengths are the analysis of the chemical composition of the essential oil and the methods of evaluation of biological activity in addition to the in vitro method with DPPH. The results are useful in the context of the need for natural preservatives. The results are interesting and the study is very well written, clear and concise.
Specific comments:
L41 and L 188 P. apterus, please specify the species.
In Tables 4-11, I did not understand if the comparison is made on the column or row, I guess on the column. In this case, if the lowest mean value 12.31, in Table 4, is significantly different from the highest mean value 92.35 then the mean values should have different letters. Please check this.
L 306 Do you know what basil cultivar was used? If so, you should specify this.

Author Response

Reviewer #4

It is an interesting and up-to-date study of the bioactivity of basil essential oil. The strengths are the analysis of the chemical composition of the essential oil and the methods of evaluation of biological activity in addition to the in vitro method with DPPH. The results are useful in the context of the need for natural preservatives. The results are interesting, and the study is very well written, clear and concise.

Thank you very much for the positive evaluation and for your time.

Point 1: Specific comments: L41 and L 188 P. apterus, please specify the species.

Response: It was changed.

Point 2: In Tables 4-11, I did not understand if the comparison is made on the column or row, I guess on the column.

Response: Statistically differences were evaluated in row, not column. It is mean that was compare different concentration within one species of microorganisms.

Point 3: In this case, if the lowest mean value 12.31, in Table 4, is significantly different from the highest mean value 92.35 then the mean values should have different letters.

Response: It was checked and recalculate, and letters are right.

Point 4: Please check this. L 306 Do you know what basil cultivar was used? If so, you should specify this.

Response: It is commercial produced essential oils. Producer just describe Ocimum basilicum plants from Vietnam.

Round 2

Reviewer 1 Report

4.1 section

Which apparatus did you use? How long did the steam distillation last? 

How much did you extracted (amount of fresh plant)?

Molecules 202126(20), 6157; https://doi.org/10.3390/molecules26206157

Which chromatographic condition did you use? 

Author Response

Reviewer #1

4.1 section

Which apparatus did you use? How long did the steam distillation last? 

How much did you extracted (amount of fresh plant)?

Molecules 202126(20), 6157; https://doi.org/10.3390/molecules26206157

Which chromatographic condition did you use? 

Response: In section 4.1. is described that essential oil Ocimum basilicum was purchased form Hanus Ltd. company. Essential oil was commercial, we didn’t extract and distillate in laboratory.

Chromatographic condition of essential oil was described in manuscript part 4.2.

Thank you very much for constructive comments.

Round 3

Reviewer 1 Report

There are several grammar errors in the main text. Here some of them. 

Line 323: did you mean 1 μL as injection volume?

Lines 317-320: correct the sentence

Line 331: correct the verb. 

I recommend the manuscript for publication in Plants.